

# LincRNA00612 inhibits apoptosis and inflammation in LPS-induced BEAS-2B cells via enhancing interaction between p-STAT3 and A2M promoter

Xinru Xiao[1,2,*], Wei Cai[1,*], Ziqi Ding[1], Zhengdao Mao[1], Yujia Shi[1] and Qian Zhang[1]

[1] Department of Respiratory and Critical Care Medicine, The Affiliated Changzhou No. 2 People's Hospital of Nanjing Medical University, Changzhou, Jiangsu, China
[2] Department of the Second Clinical College, Dalian Medical University, Dalian, Liaoning, China
* These authors contributed equally to this work.

Corresponding author
Qian Zhang,
qianzhang@njmu.edu.cn

## ABSTRACT

Long non-coding RNAs (lncRNAs) have been reported as key regulators of chronic obstructive pulmonary disease (COPD). This study aimed to figure out the regulatory mechanism as well as the effects of lncRNA00612 (LINC00612) in lipopolysaccharide (LPS)-induced inflammation and apoptosis in BEAS-2B cells. LINC00612 and its co-expressed gene alpha-2-macroglobulin (A2M) were strikingly downregulated in the peripheral venous blood of COPD patients. Overexpressed LINC00612 enhances BEAS-2B cells against apoptosis and inflammatory reactions mediated by LPS, however, an A2M knockdown can attenuate the degree of the enhancement. Bioinformatics analysis revealed putative binding sites between LINC00612, signal transducer and activator of transcription 3 (STAT3) and the A2M promoter, while RNA antisense purification and Chromatin immunoprecipitation were performed to confirm the prediction. Knockdown of LINC00612 impaired the binding of p-STAT3 to the promoter of A2M, which meant that LINC00612 was critical for the binding of STAT3 with the A2M promoter. Therefore, it can be concluded that LINC00612 ameliorates LPS-induced cell apoptosis and inflammation *via* recruiting STAT3 to bind to A2M. This conclusion will serve as a theoretical foundation for the treatment of COPD.

## INTRODUCTION

Chronic obstructive pulmonary disease (COPD), a progressive degenerative disease involved in the respiratory system, is a common reason for death all over the world (*Barnes, 2018*). A thorough pathogenesis of the disease is still on the way. Cell apoptosis, epithelial to mesenchyme transition, oxidative stress, inflammation, and protease/anti-protease imbalance are allegedly the key cause of lung destruction under COPD (*Wang et al., 2020b*). Hence, it is essential to further elucidate the specific molecular mechanism of COPD.

Long non-coding RNAs (lncRNAs) are non-coding RNAs, which is longer than 200 nucleotides and lacking protein-coding capacity (*Mercer, Dinger & Mattick, 2009*). It is increasingly proven that different biological progress has involved lncRNAs; for instance, epigenetic modification, control of transcription, translation, mRNA splicing, and genomic imprinting (*Quinn & Chang, 2016*). *Luo et al. (2020)* found that a non-coding RNA named LINC00612, dropped in COPD tissues and could regulate inflammation, oxidative stress, and apoptosis in human pulmonary microvascular endothelial cells *via* targeting miR-31-5p/NOTCH1 axis. Interestingly, our previous studies analyzed and identified dysregulation of LINC00612 in patients with COPD, and LINC00612 was co-expressed with alpha-2-macroglobulin (A2M) (*Qian et al., 2018*).

A2M is an acute-phase protein (*Poznanović, Petrović & Magić, 1997*). When reacting with proteinases, A2M is transformed and hence becomes able to bind to cytokines, growth factors, and cellular receptors (*Borth, 1992*; *Vandooren & Itoh, 2021*). Thus, A2M can be important in maintaining the proteinase-mediated homeostasis of cytokines and growth factors (*Borth, 1992*; *Cuéllar, Cuéllar & Scuderi, 2016*). Additionally, an anti-apoptotic behavior of A2M is also reported (*Lee & Piedrahita, 2002*; *Liu et al., 2019*). The synthesis of A2M is primarily regulated at the transcriptional level. It is mediated by pro-inflammatory cytokines, mainly interleukin-6 (IL-6) and the regulatory proteins which are under its control (*Poli, 1998*; *Kunz et al., 1989*). Signal transducer and activator of transcription (STAT) family are the major participants in the IL-6 signaling cascade (*Alonzi et al., 2001*; *Heinrich et al., 2003*). STAT3, a member of the STAT family, is a transcription factor and an intracellular signal sensor activated by cytokines, growth factors, and tyrosine kinases (*Darnell, 1997*; *Levy & Lee, 2002*). After binding cytokines to their cognate receptors, cytokine receptor-associated protein-tyrosine kinases of the Janus kinase (JAK) family phosphorylate STAT3 protein. The phosphorylated STAT3 then forms dimers *via* their SH2 domains and rapidly translocated from cytoplasm to nucleus, where it binds regulatory DNA elements of target genes (*Johnson, O'Keefe & Grandis, 2018*). From previous studies, we know that activated STAT3 by other transacting proteins could promote A2M gene transcription (*Ripperger et al., 1995*; *Levy & Darnell, 2002*; *Uskokovic et al., 2007*).

Therefore, we hypothesized that LINC00612, STAT3, and A2M played an important role in the occurrence and development of COPD. In the present study, we were committed to elucidating the regulatory relationship between LINC00612, STAT3, and A2M and their specific role in the pathogenesis of COPD.

## MATERIALS AND METHODS

### Patient specimens

We obtained the clinical samples of peripheral venous blood in the interval from January 2022 to July 2022 from Changzhou Second People's Hospital. The sample involves the healthy control group ($n = 34$) and the COPD group ($n = 32$). All subjects underwent routine renal function, liver function, blood tests, blood lipids, blood glucose, and electrocardiogram to exclude underlying diseases. No one had bronchiectasis, tuberculosis, asthma, or other confounding inflammatory diseases or took corticosteroids, antibiotics, or

**Table 1 Sequences of small inferring RNAs.**

|  | Sense | Antisense |
|---|---|---|
| siNC | UUCUCCGAACGUGUCACGUTT | ACGUGACACGUUCGGAGAATT |
| siLINC00612#1 | GCCAUGUUGAAAGUUGGUATT | UACCAACUUUCAACAUGGCTT |
| siLINC00612#2 | GCUCCCUAUGUGAGCCAUUTT | AAUGGCUCACAUAGGGAGCTT |
| siA2M#1 | GCCUAUACACAUAUGGGAATT | UUCCCAUAUGUGUAUAGGCTT |
| siA2M#2 | CCCUUUCACCGUGGAGGAATT | UUCCUCCACGGUGAAAGGGTT |
| siSTAT3#1 | CCCGGAAAUUUAACAUUCUTT | AGAAUGUUAAAUUUCCGGGTT |
| siSTAT3#2 | CCACUUUGGUGUUUCAUAATT | UUAUGAAACACCAAAGUGGTT |

medications for extrapulmonary diseases before taking the experiment. For the following study, we kept the samples at −80 °C. This study was endorsed by the ethics institute of Affiliated Changzhou No. 2 People's Hospital of Nanjing Medical University (2022KY113-01) and acquired informed consent from all participants.

## Cell cultivation and treatment of lipopolysaccharide (LPS)

We purchased healthy lung epithelial cell lines for humans (BEAS-2B) from Zhongqiaoxinzhou Biotech (Shanghai, China). The cultivation of these cells was conducted in a humidified atmosphere at 37 °C with 5% $CO_2$ in Dulbecco's modified Eagle medium (DMEM; Gibco, Carlsbad, CA, USA). Moreover, 10% fetal bovine serum and 1% penicillin/streptomycin were added to DMEM. To construct a model of COPD *in vitro*, we incubated BEAS-2B cells with increasing doses of Lipopolysaccharide (LPS) for 24 h.

## Cell transfection

The overexpression plasmid of LINC00612 (LINC00612), the small interfering RNAs against LINC00612 (siLINC00612), STAT3 (siSTAT3), and A2M (siA2M), and control groups (vector and siNC) were synthesized by GenePharma (Suzhou, China). Cell transfection was carried out with Lipofectamine 3000 (Invitrogen, Carlsbad, CA, USA). Cells were harvested and extracted for further study 48 h after the transfection. Sequences used in this part are presented in Table 1.

## Real-time quantitative polymerase chain reaction (RT-qPCR)

TRIzol reagent (Invitrogen, Carlsbad, CA, USA) was used to extract total RNA from samples or cells. Using Cytoplasmic & Nuclear RNA Purification Kit (Norgen Biotek, Thorold, ON, Canada) according to the manufacturer's instructions, we extracted nuclear and cytoplasmic RNA. With the help of the manufacturer's instructions, we used the First Strand cDNA Synthesis Kit (Vazyme, Nanjing, China) to synthesize the first strand of complementary DNA. Afterward, we conducted qRT-PCR (ABI, Foster City, CA, USA) to assess the degree of RNA expression using AceQ qPCR SYBR Green Master Mix (Vazyme, Nanjing, China). We regarded β-actin as the control for the transcript of cytoplasm and a housekeeping gene for LINC00612, A2M, and STAT3, and U6 as the control for the transcript of the nuclei. We assessed the result using the $2^{-\Delta\Delta Ct}$ method. Here are the primers: LINC00612 (sense, 5′-CCCCTGATGTACGCCTGTTT-3′; antisense, 5′-CCAA

CACATGGCTCTGCCTA-3′); A2M (sense, 5′-AGGAAATCGCATCGCACAATG-3′; antisense, 5′-ACGGTGAAAGGGTGCTCTG-3′); STAT3 (sense, 5′-AGAAGGACATCA GCGGTAAGA-3′; antisense, 5′-GGATAGAGATAGACCAGTGGAGAC-3′); U6 (sense, 5′-CGCTTCGGCAGCACATATAC-3′; antisense, 5′-TTCACGAATTTGCGTGTCA TC-3′); β-actin (sense, 5′-CGTGGACATCCGCAAAGA-3′; antisense, 5′-GAAGGTGGA CAGCGAGGC-3′).

## Assay of cell counting kit-8 (CCK-8)

With CCK-8 assay kits (Dojindo, Tokyo, Japan), a determination of cell viability was made. We seeded transfected BEAS-2B cells ($3 \times 10^3$/well) into plates each having 96 wells and altogether having 100 µL of the medium. We added a total of 10 µL CCK-8 solutions (Dojindo, Tokyo, Japan) 24 h after diverse treatments, and then we incubated the cells in an atmosphere whose temperature remained 37 °C and had 5% of $CO_2$ for 2 h. With the help of an enzyme-labelled instrument (Thermo, Waltham, MA, USA), in every well we measured the absorbance at 450 nm.

## Assay of flow cytometry

For the assay of apoptosis, the Annexin V-FITC/PI Apoptosis Detection Kits (Vazyme, Nanjing, China) was used. We gathered BEAS-2B cells with trypsin and then washed the cells twice with ice-cold phosphate saline buffer (PBS), and centrifuged the cells at 1,000 g for 5 min. Then, we resuspended the cells in 100 µL binding buffer and incubated with 5 µL of propidium iodide and 5 µL of Annexin V-FITC in dark conditions for 10 min. Afterward, we analyzed the cells using a flow cytometer (Becton Dickinson and Co, Franklin Lakes, NJ, USA). Proportions (%) of dead cells or those apoptosis cells were adopted as rates of apoptosis.

## Enzyme-linked immunosorbent assay (ELISA)

We employed ELISA kit (Beyotime, Shanghai, China) to calculate the levels of IL-6 and TNF-α. In other words, we centrifuged the supernatant at 3,000 rpm for 5 min. We then added the sample and standard substance respectively into the reaction well and incubated them for 2 h at room temperature. Subsequently, we used the scrubbing solution to wash the plate and placed biotinylated anti-TNF-α (Beyotime, Shanghai, China) or anti-IL-6 (Beyotime, Shanghai, China) into well for 1 h. We incubated the sample with streptavidin-marked horseradish peroxidase (Beyotime, Shanghai, China) for 20 min. Further, we introduced the solution and figured out the amount of IL-6 and TNF-α by using a microplate reader (Thermo, Waltham, MA, USA) to measure the absorbance (450 nm).

## RNA antisense purification (RAP)

We designed and synthesized five RAP probes targeting LINC00612, and each DNA oligonucleotide probe was modified with a 5′ biotin. In Table 2, we manifested the sequences of the probes. We employed the RAP Kit (BersinBio, Guangzhou, China) to perform the RAP. By the instructions of the manufacturer, around $4 \times 10^7$ cells were crosslinked with 1% formaldehyde. Subsequently, we dissolved the cross-linked cells with a

**Table 2** Biotin-labeled LINC00612 probe sequences for RAP assay.

|  | Probe sequence |
|---|---|
| Probe#1 | TGTCCTGCTCACGTGGCTAGATCTTTCAAGCTATTTCACA |
| Probe#2 | TCCCAGGAAGCTTGCGATGAGGAATGAGGATGTAAAAT |
| Probe#3 | AAATATTCAGGAAACAGGCGTACATCAGGGGCTGTTCTTT |
| Probe#4 | TAATGGCTCACATAGGGAGCAGTTGGAGGGTGATGTGGAT |
| Probe#5 | GATTCCAGCCAAGTCTTGTTTATAGACACA |

lysis buffer with inhibitors of protease and RNase. After eliminating DNA using Dnase, we incubated the lysates with the probe mixture or the control probes of LINC00612 and then immobilized them with streptavidin-coated magnetic beads. After washing beads that had captured hybrids three times using a washing buffer, we eluted protein and RNA respectively from magnetic beads. In the end, we isolated RNA and subjected it to assays of qPCR. Protein was isolated and subjected to western blot.

## Chromatin immunoprecipitation (CHIP)

We conducted ChIP assay with a commercially available Enzymatic Chromatin IP kit (BersinBio, Guangzhou, China). The manufacturer instructed that approximately two million BEAS-2B cells should be crosslinked in 1% formaldehyde at room temperature for 10 min and be quenched with glycine (125 mM final concentration) for 5 min. We sheared the chromatin into 200–600 bp fragments with sonication, and handle them with RNase A (0.2 mg/mL) at 37 °C for 30 min. We incubated 3 mg of anti-phosphorylated STAT3 (Abcam, Cambridge, UK) or IgG (BersinBio, Guangzhou, China) with chromatin at 4 °C for the whole night. Using Dynal-beads Protein G (BersinBio, Guangzhou, China), we recovered protein-antibody complexes. Then, we isolated DNA using phenol/chloroform, and performed quantitative PCR with three primers designed according to the binding site of STAT3 and A2M promoter by Primer Premier 5: primer#1 (sense, 5′-TCCTGGATCACCTCTTTCTAGC-3′; antisense, 5′-GGTATTCAGAGGAGTACTCAGTG-3′); primer#2 (sense, 5′-TAGTTTTCATTTCATGCATGTGC-3′; antisense, 5′-CTGAAGTATGAGTAGTGGTTGTC-3′); primer#3 (sense, 5′-TCCCACCAATCTGTGTTTCTTC-3′; antisense, 5′-GTGCTCAGTGAATATATCAATGTG-3′).

## Western blot assay

After centrifuging cells at 12,000 g for 10 min at 4 °C, we lysed them in radio-immunoprecipitation assay buffer (Beyotime, Shanghai, China) with protease and phosphatase inhibitors (Beyotime, Shanghai, China) for 30 min under the same temperature. With the help of BCA Protein Assay Kits (Beyotime, Shanghai, China), we determined the protein contents. Then protein samples were subjected to 10% sodium dodecyl sulfate-polyacrylamide gel electrophoresis (SDS-PAGE) and transblotted onto polyvinylidene difluoride (PVDF) membranes. Using the 5% skimmed milk, we blocked

the antigen binding that was not specific for one and a half hours, then we cultured overnight the membranes with primary antibodies against A2M, total STAT3, phosphorylated STAT3, cleaved-caspase3, pro-caspase3, GAPDH (All from Abcam, Cambridge, UK) at 4 °C. After washing the membranes with Tris Buffer Saline Tween (TBST) three times and cultivated on them with goat anti-rabbit IgG secondary antibody (Abcam, Cambridge, UK) at room temperature for 1.5 h, we washed them again with TBST. In the end, we visualized the proteins and quantified them with an enhanced chemiluminescence kit (Tanon, Shanghai, China). We evaluated the proteins with densitometry by the software named ImageJ (National Institutes of Health, Bethesda, MD, USA).

### Bioinformatics analysis

We used the catRAPID (http://s.tartaglialab.com/page/catrapid_group) in predicting the binding of LINC00612 to transcription factors STAT3, and JASPAR (http://jaspar.genereg.net/) in the prediction of the binding region of STAT3 to the A2M promoter sequence.

### Statistical analysis

GraphPad Prism v.8.0 (GraphPad Software Inc., San Diego, CA, USA) was used in the analysis of all data. We afterward presented the results as the means ± standard error of the mean (SEM) after three independent trials. In seeking the differences between the groups, Student's $t$-tests and the one-way analysis of variance (ANOVA) were implemented. Associations between LINC00612 and A2M expression in COPD patients were analyzed using Pearson's correlation analysis. The receiver-operating characteristic (ROC) curve was adopted to examine the value of LINC00612 in the diagnosis of COPD. Values with $P < 0.05$ were considered to indicate a statistically significant difference.

## RESULTS

### LINC00612 and A2M was downregulated in COPD patients and LPS-induced BEAS-2B cells

Our previous study demonstrated that LINC00612 was co-expressed with A2M (*Qian et al., 2018*). To reveal the correlation between LINC00612 and A2M, RT-qPCR was performed and data showed that LINC00612 and A2M expressions were significantly downregulated in the peripheral venous blood of COPD patients when compared to normal control (Figs. 1A and 1B). Table 3 presents the main clinical data of all participants. There was no difference in gender, age, BMI, smoking history and eosinophils between the two groups. The $FEV_1/FVC$ and $FEV_1\%$ predicted of COPD patients were significantly lower than controls. To assess the LINC00612 value in diagnosing COPD, we graphed the ROC curve. It is revealed then that LINC00612 could distinguish COPD patients from healthy ones with areas under the curve (AUC) of 0.9292 (95% CI [0.8703–0.9882]; $P < 0.0001$) (Fig. 1C). Furthermore, Pearson's correlation analysis revealed a positive correlation relationship between LINC00612 and A2M in the peripheral venous blood of COPD patients ($r = 0.5017$, $P < 0.01$) (Fig. 1D). Subsequently, we confirmed in BEAS-2B cells the effect the treatment of LPS had on the viability of cells to assess the feasibility of

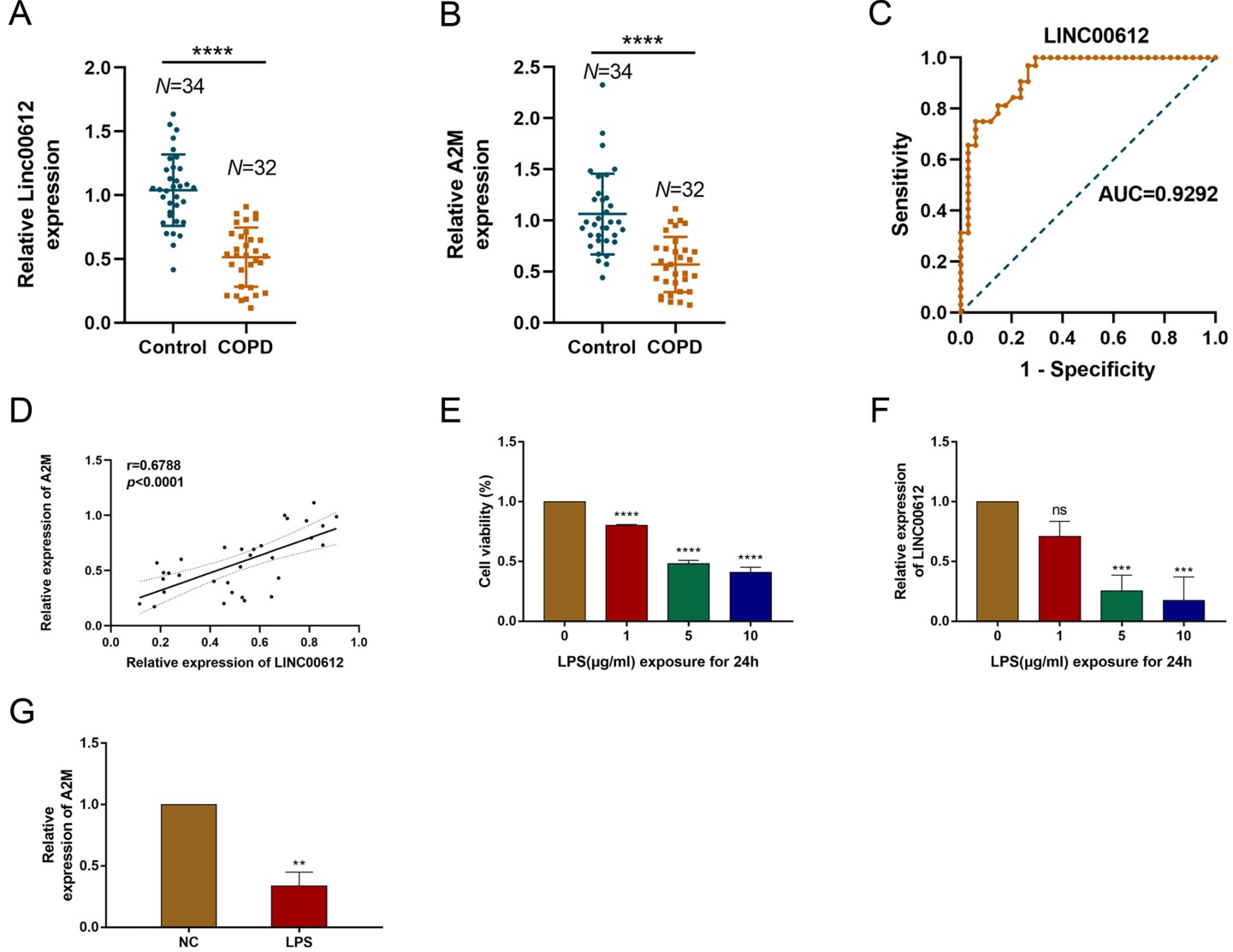

**Figure 1 The expression level of LINC00612 and A2M in COPD patients and LPS-induced BEAS-2B cells.** (A and B) RT-qPCR assay was applied to measure the expression level of LINC00612 and A2M in the peripheral venous blood of COPD patients ($n$ = 32) compared to the control subjects ($n$ = 34). (C) ROC curve was used to investigate the value of LINC00612 in COPD diagnosis. (D) Pearson's correlation analysis was employed to reveal the relationship between LINC00612 expression and A2M expression. (E) CCK-8 assay was performed to illustrate the impacts of LPS on cell viability in BEAS-2B cells. (F) The expression level of LINC00612 was assessed by RT-qPCR assay in BEAS-2B cells induced by LPS at different concentrations for 24 h. (G) The expression level of A2M was assessed by RT-qPCR assay in BEAS-2B cells induced by LPS at 10 μg/mL for 24 h. Data were expressed as mean ± SEM and from three independent experiments. ns, no significant ($P > 0.05$), **$P < 0.01$, ***$P < 0.001$, ****$P < 0.0001$.

the LPS-induced COPD model. When different concentrations of LPS (1, 5 and 10 μg/mL) were used to treat BEAS-2B cells for 24 h, cell viability was significantly reduced (Fig. 1E). In addition, LINC00612 expression was reduced by LPS in a concentration-dependent manner (Fig. 1F). When BEAS-2B cells were treated with 10 μg/mL LPS for 24 h, LINC00612 expression was lowest. Accordingly, treatment with 10 μg/mL LPS also induced the downregulation of A2M in BEAS-2B cells (Fig. 1G). These results confirm the

**Table 3 Characteristics of patients.**

| Variable | Control ($n$ = 34) | COPD ($n$ = 32) | Statistic | $P$ value |
|---|---|---|---|---|
| Gender (female/male) | 7/27 | 6/26 | $X^2$ = 0.035 | 0.851 |
| Age (years) | 65.97 ± 5.89 | 68.31 ± 9.92 | t = −1.157 | 0.253 |
| BMI (kg/m$^2$) | 25.65 ± 3.34 | 24.20 ± 3.26 | t = 1.772 | 0.081 |
| Smoking history (no/yes) | 7/27 | 8/24 | $X^2$ = 0.183 | 0.669 |
| FEV$_1$/FVC | 82.49 ± 5.90 | 45.60 ± 11.62 | t = 16.111 | <0.001 |
| FEV$_1$%pre | 98.45 ± 16.99 | 52.03 ± 18.64 | t = 10.585 | <0.001 |
| GOLD stage | | | | |
| 1 | | 4 | | |
| 2 | | 12 | | |
| 3 | | 14 | | |
| 4 | | 2 | | |
| Eosinophils (×10$^9$/L) | 0.15 ± 0.16 | 0.21 ± 0.19 | t = −1.504 | 0.138 |

**Note:**
  Data were presented as mean ± SEM.

viability of the LPS-induced COPD model. Therefore, in subsequent studies on LPS, we pretreated the cells with 10 µg/mL LPS for 24 h before various treatments.

## Overexpression of LINC00612 weakened the LPS-induced effects of apoptosis and inflammation in BEAS-2B cells

Using the LPS-induced COPD model, we investigated the effect of LINC00612 in COPD pathogenesis. LINC00612 was upregulated in BEAS-2B cells after being transfected with LINC00612 overexpression plasmid (Fig. 2A). CCK-8 assay indicated that there is a repression of the cell viability in LPS-induced BEAS-2B cells, while the overexpression of LINC00612 reinstated the degree (Fig. 2B). Flow cytometer assay demonstrated that cell apoptosis was exacerbated in LPS-induced BEAS-2B cells, nevertheless, the overexpression of LINC00612 also relieved the effect (Fig. 2C). Data also showed the ratio of apoptosis-related proteins cleaved-caspase3 and pro-caspase3 had increased in LPS-induced BEAS-2B cells, but LINC00612 overexpression restored these changes (Fig. 2D). ELISA demonstrated that pro-inflammatory cytokines IL-6 and TNF-α levels were upregulated in LPS-induced BEAS-2B cells and the overexpression of LINC00612 attenuated the upregulation (Figs. 2E and 2F). The results sketched above suggested that LPS-induced apoptosis and inflammation in BEAS-2B cells can be abated or even eliminated by the upregulation of LINC00612, suggesting that LINC00612 is a crucial contributor to the development of COPD.

## A2M was positively regulated by LINC00612

To reveal whether A2M was regulated by LINC00612, we first determined the transfection efficiency of small interfering RNAs against LINC00612 and A2M (Figs. 3A and 3B). And siLINC00612#2 and siA2M#2 were chosen for subsequent study owing to their most successful interfering efficiency. Subsequently, the expression level of A2M in BEAS-2B cells transfected with vector, LINC00612 plasmid, siNC and siLINC00612 was detected by

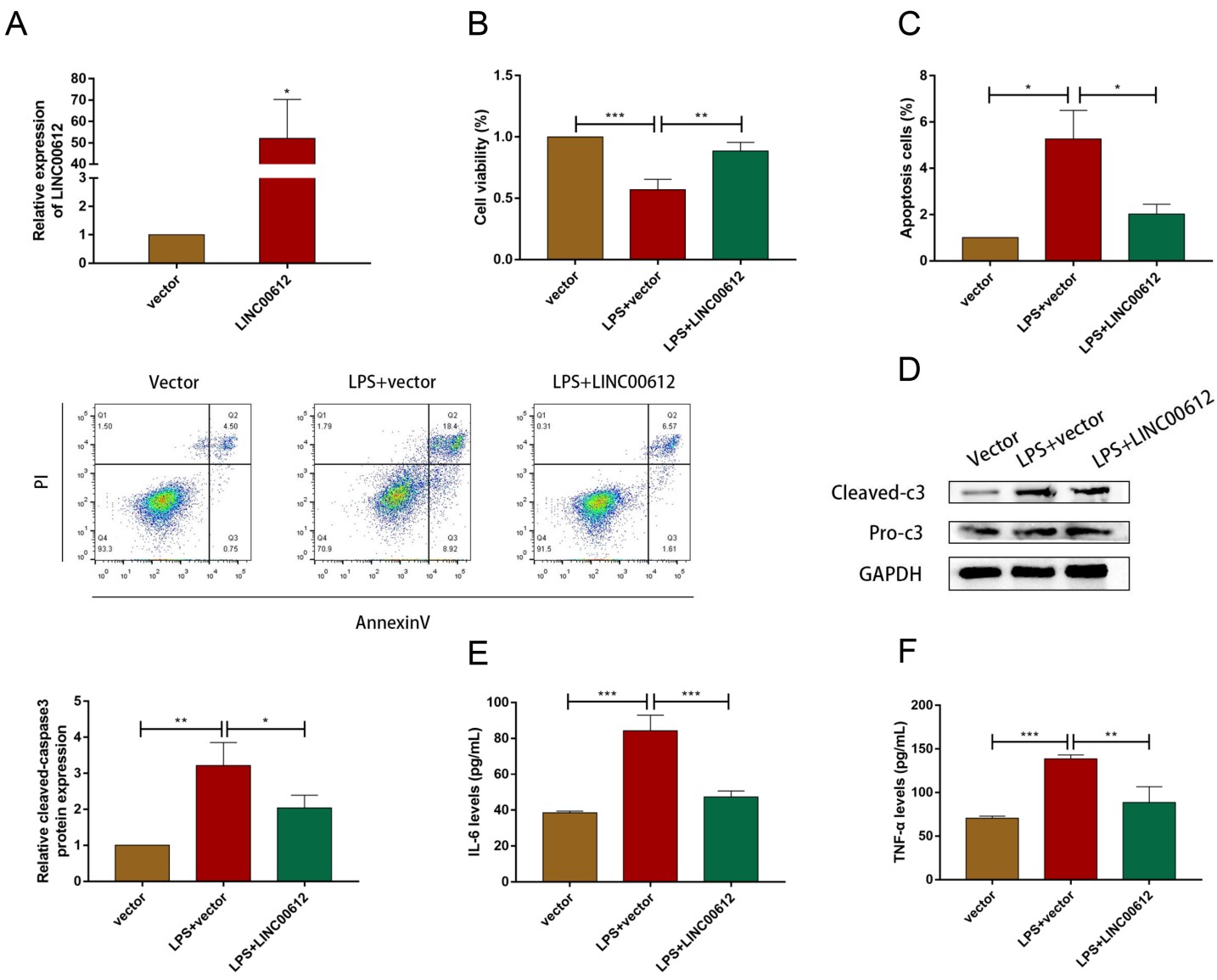

**Figure 2 LINC00612 regulated apoptosis and inflammation in BEAS-2B cells treated with LPS.** (A) The expression level of LINC00612 was quantified by RT-qPCR in BEAS-2B cells transfected with LINC00612 or vector. (B–F) BEAS-2B cells were divided into three groups: vector, LPS + vector, and LPS+LINC00612. (B) CCK-8 assay was performed to illustrate the impacts between LPS and LINC00612 on cell viability in BEAS-2B cells. (C) The percentage of apoptotic cells was shown by flow cytometry assay in BEAS-2B cells. (D) The protein expression level of cleaved-caspase3 and pro-caspase3 was measured by western blot assay in BEAS-2B cells. (E and F) The expression levels of IL-6 and TNF-α were assessed by ELISA kits in supernatant. Data were expressed as mean ± SEM and from three independent experiments. $*P < 0.05$, $**P < 0.01$, $***P < 0.001$.

RT-qRCR and western blot. The results showed that the mRNA and protein expression of A2M increased after LINC00612 overexpression and decreased after LINC00612 knockdown (Figs. 3C and 3D). A2M knockdown reversed the role of LINC00612 overexpression (Figs. 3E and 3F). All these results suggested that A2M was positively regulated by LINC00612.

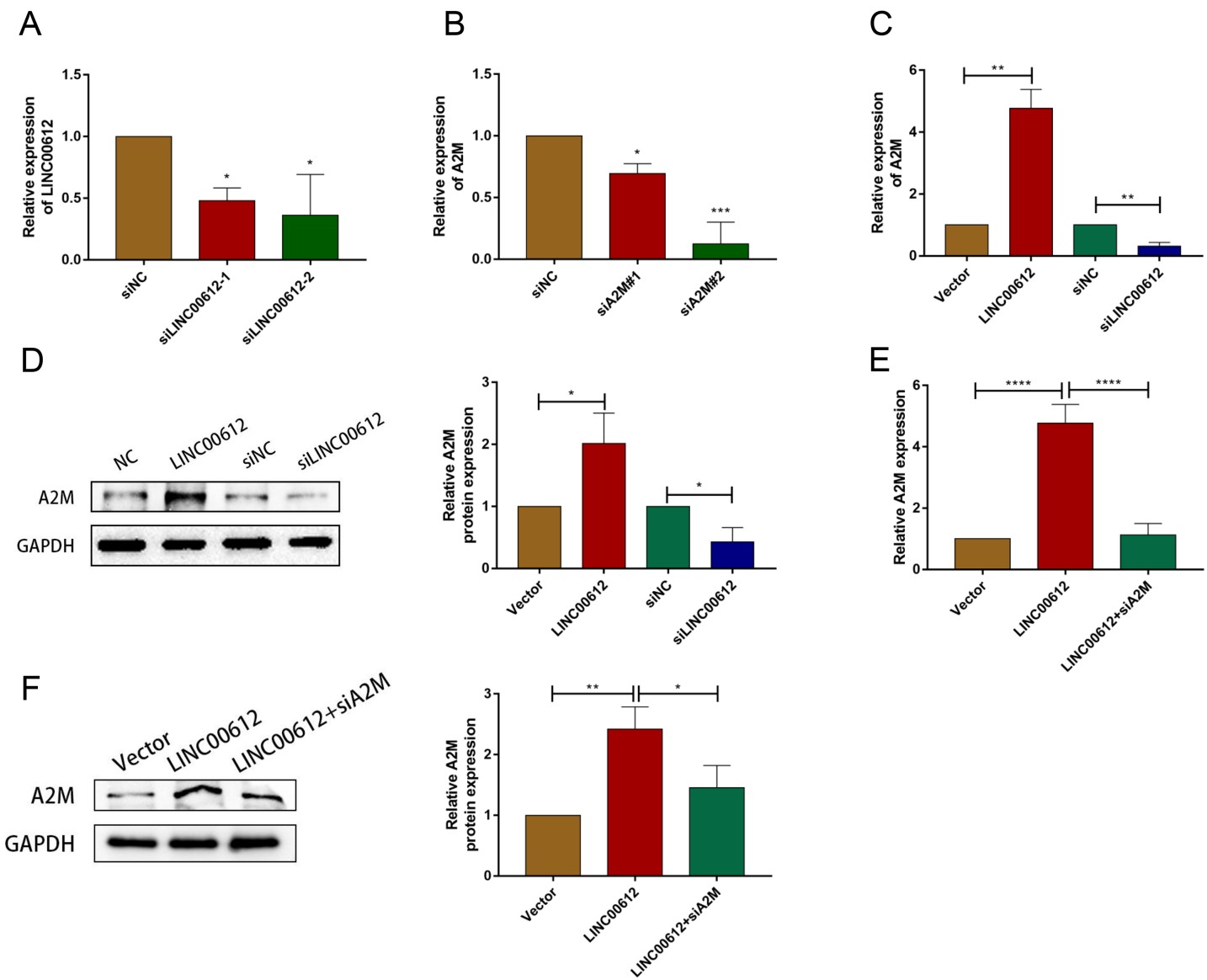

**Figure 3 A2M was regulated by LINC00612.** (A) The knockdown efficiency of si-LINC00612#1 and si-LINC00612#2 was determined by detecting the mRNA levels of LINC00612 *via* RT-qPCR. (B) The knockdown efficiency of si-A2M#1 and si-A2M#2 was determined by detecting the mRNA levels of A2M *via* RT-qPCR. (C and D) The mRNA and protein expression level of A2M was measured by RT-qRCR and western blot assay in BEAS-2B cells transfected with vector, LINC00612 plasmid, siNC, siLINC00612. (E and F) The mRNA and protein expression level of A2M was measured by RT-qRCR and western blot assay in LPS-induced BEAS-2B cells transfected with vector, LINC00612 plasmid, and LINC00612 co-transfected with siA2M. $^*P < 0.05$, $^{**}P < 0.01$, $^{***}P < 0.001$, $^{****}P < 0.0001$.

## A2M knockdown prevented the inhibitory effects of LINC00612 on LPS-induced apoptosis and inflammation in BEAS-2B cells

We revealed the effects between LINC00612 and A2M knockdown on apoptosis and inflammation induced by LPS to figure out whether the physiological process of COPD was mediated by LINC00612 with the modulation of A2M. As shown in Figs. 4A and 4B, LINC00612 overexpression improved the changes of cell viability and apoptosis induced by LPS but the effect was eliminated by the silencing of A2M. At the same time, it is

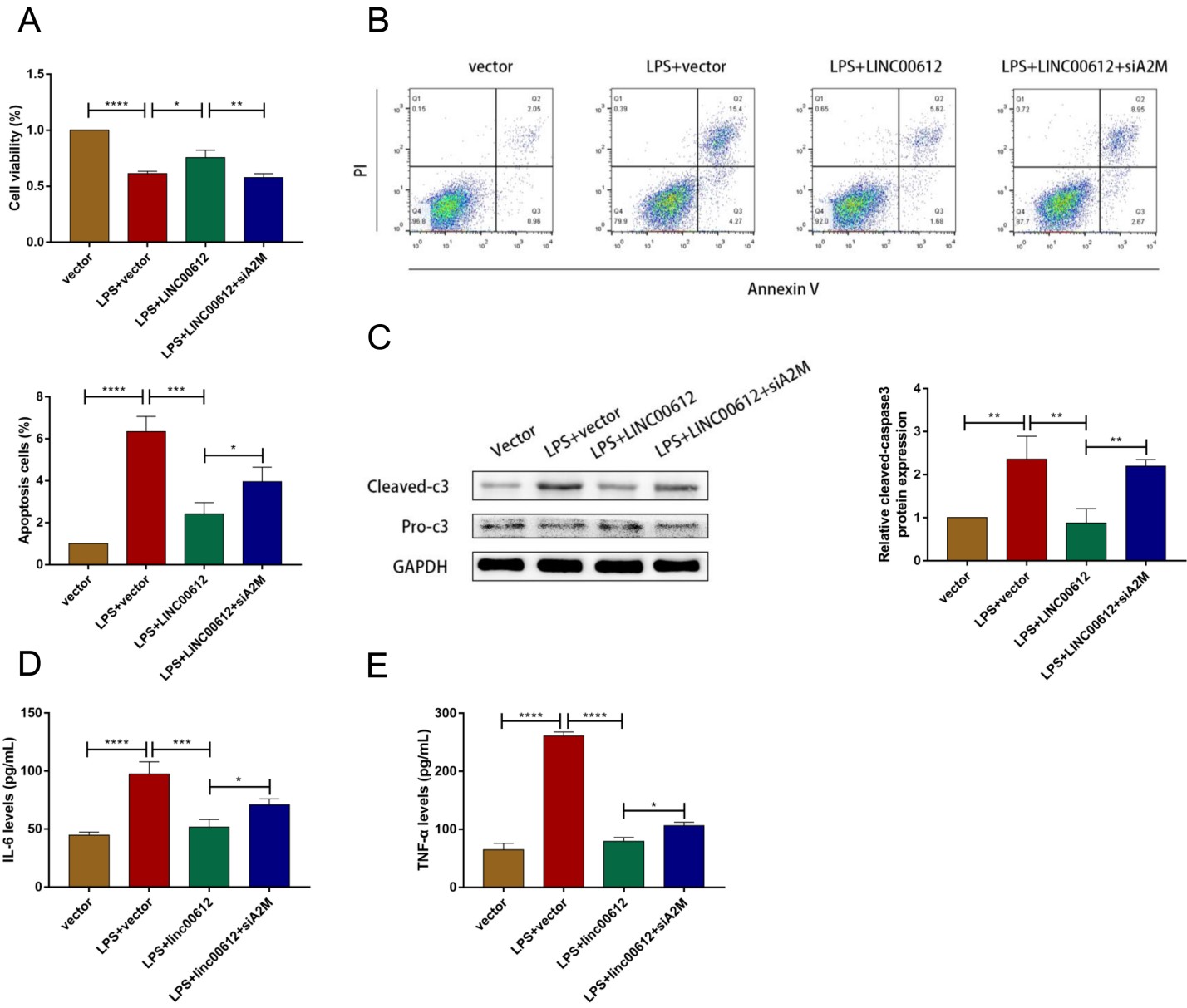

**Figure 4 LINC00612 ameliorated LPS-induced disorders by regulating A2M.** (A–E) BEAS-2B cells were divided into four groups: vector, LPS + vector, LPS+LINC00612, LPS+LINC00612+siA2M. (A) CCK-8 assay was employed to illustrate the impacts between LINC00612 and A2M on cell viability in LPS-induced BEAS-2B cells. (B) The percentage of apoptotic cells was shown by flow cytometry assay in BEAS-2B cells. (C) The protein expression level of cleaved-caspase3 and pro-caspase3 was measured by western blot assay in BEAS-2B cells. (D and E) The expression levels of IL-6 and TNF-α were assessed by ELISA kits in supernatant. Data were expressed as mean ± SEM and from three independent experiments. *$P < 0.05$, **$P < 0.01$, ***$P < 0.001$, ****$P < 0.0001$.

revealed by the result that the ratio between cleaved-caspase3 and pro-caspase3 was largely decreased by LINC00612, while A2M silencing impeded these influences (Fig. 4C). Moreover, the generation of IL-6 and TNF-α in LPS-induced BEAS-2B cells was blocked by LINC00612, which was inhibited by A2M knockdown (Figs. 4D and 4E). Hence, by regulating A2M, LINC00612 protected BEAS-2B cells against LPS-induced apoptosis and inflammation.

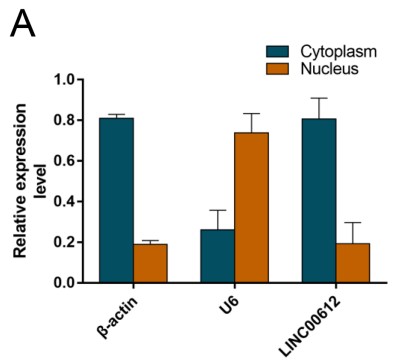
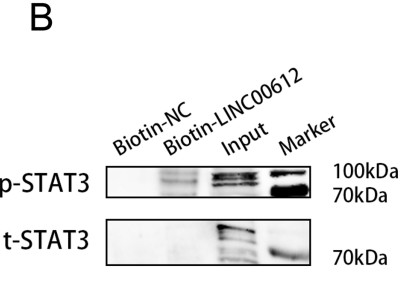
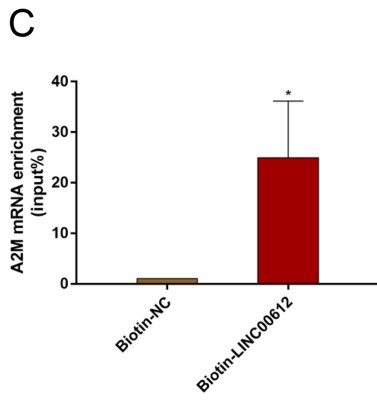

**Figure 5 LINC00612 directly interacted with phosphorylated STAT3.** (A) RT-qPCR was performed in separated cytoplasm RNA and nuclear RNA isolated from BEAS-2B cells. (B and C) RAP was conducted using biotin-labeled RNA probes targeting LINC00612. (B) The protein expression level of p-STAT3 was measured by western blot assay in biotin-labeled LINC00612 pull-down samples. (C) qPCR was used to measure the enrichment of A2M mRNA in biotin-labeled LINC00612 pull-down samples. Data were expressed as mean ± SEM and from three independent experiments. *$P < 0.05$.

## LINC00612 directly interacted with p-STAT3 (Tyr705)

We wanted to further elucidate how LINC00612 modulated A2M transcription. RT-qPCR was performed in separated cytoplasm RNA and nuclear RNA isolated from BEAS-2B cells, and the results indicated that LINC00612 was located in both nucleus and cytoplasm of BEAS-2B cells (Fig. 5A). As the synthesis of A2M is primarily regulated at the transcriptional level, we hypothesized that LINC00612 could regulate A2M at the transcriptional level. Although it is not fully clarified how A2M transcription is regulated, *Uskokovic et al. (2007)* found a binding site at A2M promoter for STAT3, and STAT3 promoted A2M gene transcription. With the help of prediction software analysis of catRAPID (http://s.tartaglialab.com/page/catrapid_group), we surprisingly discovered that LINC00612 had high interaction propensity with STAT3 in three segments (*i.e.*, 51 to 169, 202 to 353 and 141 to 245) (Fig. S1). Thus we conducted RAP assay with biotin-labeled RNA probes targeting LINC00612 to assess the direct engagement between LINC00612 and STAT3. Western blot with anti-p-STAT3 (Tyr705) antibody detected the existence of p-STAT3 inside pull-down samples of biotin-labeled LINC00612; however, we had not observed p-STAT3 in the control group whose samples are normal ones (Fig. 5B). These findings suggested that LINC00612 directly interacted with p-STAT3. Furthermore, we detected that compared with the negative control, A2M mRNA was significantly enriched in biotin-labeled LINC00612 pull-down samples (Fig. 5C).

## p-STAT3 Promoted A2M transcription *via* binding to A2M promoter in BEAS-2B cells

Previous study suggested that activated STAT3 increased A2M gene transcription (*Uskokovic et al., 2007*). We examined the A2M levels after STAT3 silencing in BEAS-2B cells to ascertain the causal relationship between STAT3 and A2M. We hence first determined the transfection efficiency of small interfering RNAs against STAT3; the outcome indicated that siSTAT3#1 and siSTAT3#2 repressed the expression of STAT3 at
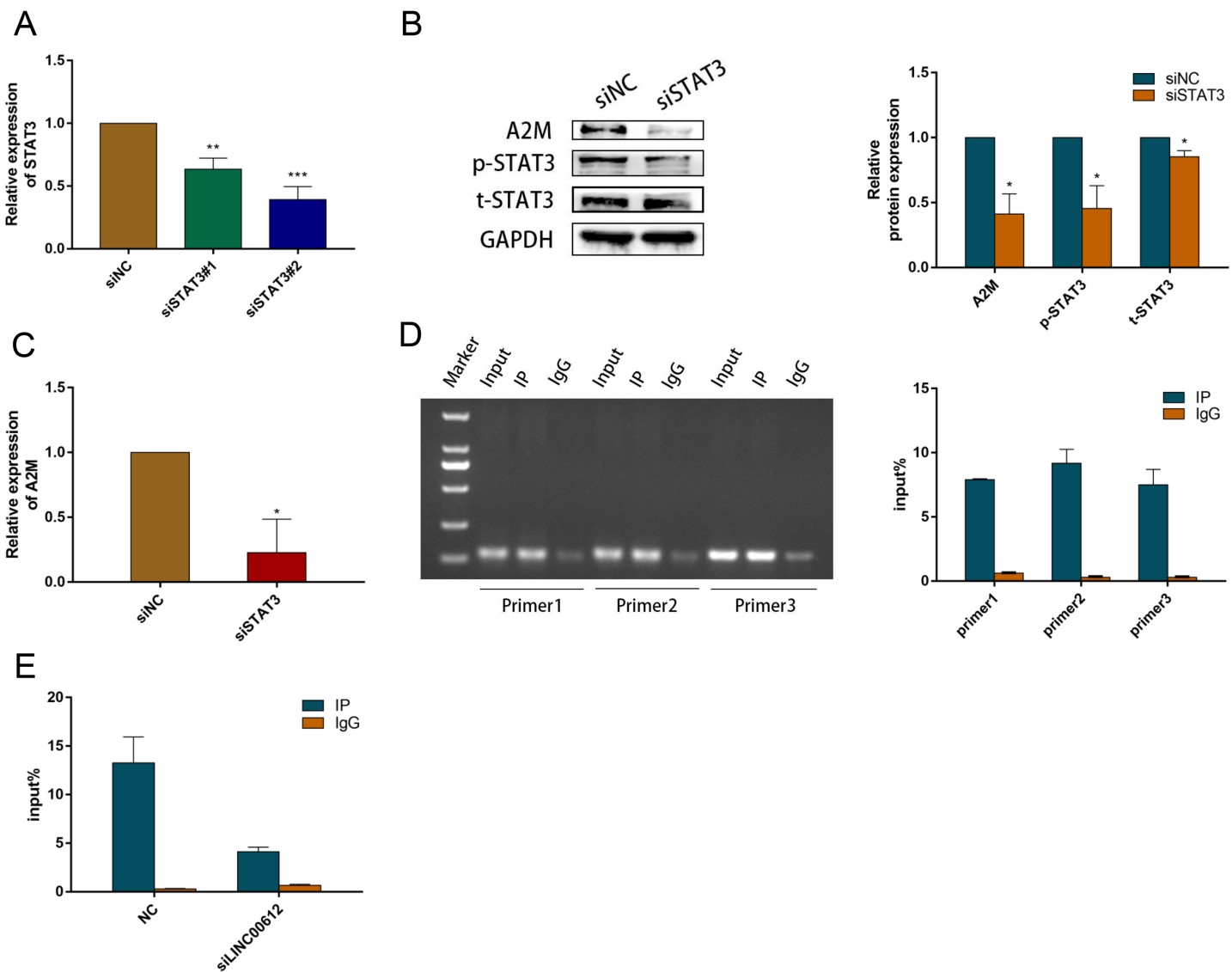

**Figure 6 LINC00612 enhancing interaction between p-STAT3 and A2M promoter.** (A) The knockdown efficiency of si-STAT3#1 and si-STAT3#2 was determined by detecting the mRNA levels of STAT3 *via* RT-qPCR. (B-C) The expression level of A2M was measured by RT-qPCR and western blot in BEAS-2B cells transfected with siNC, siSTAT3. (D) ChIP assay was performed with the p-STAT3 (Tyr705) antibody in BEAS-2B cells. (E) The enrichment of p-STAT3 (Tyr705) on A2M promoters by ChIP-PCR assay was performed in BEAS-2B cells with LINC00612 knockdown. Data were expressed as mean ± SEM and from three independent experiments. $^*P < 0.05$, $^{**}P < 0.01$, $^{***}P < 0.001$.

mRNA and protein levels in BEAS-2B cells (Figs. 6A and 6B). The protein and mRNA expression levels of A2M were inhibited by siSTAT3#2 (Fig. 6C). Previous experiments identified numerous protein-binding sites in A2M promoter region, in which two were for STAT3 (*Zhang & Darnell, 2001*; *Schaefer, Sanders & Nathans, 1995*). In addition, *Lerner et al. (2003)* found a STAT3 site (−165 to −158) and an atypical STAT3 site (−187 to −179) in the original mutation of A2M promoter by transfection analysis of reporter gene constructs. Therefore, we performed ChIP-qPCR using three PCR primers designed according to the binding site of STAT3 and A2M promoter. All three primers precipitated
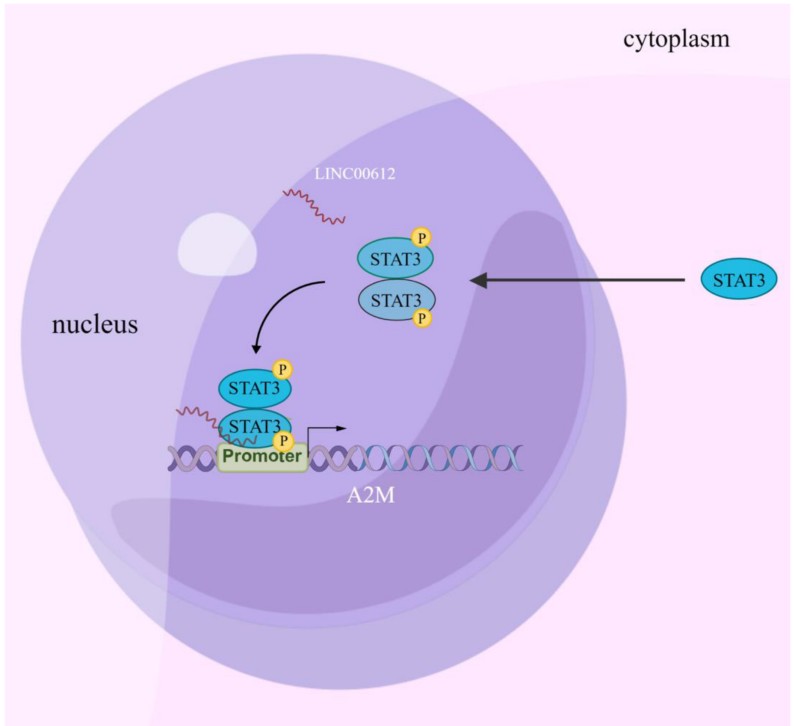

**Figure 7 Schematic diagram illustrating how LINC00612 enhances the interaction between p-STAT3 and A2M promoter to protect BEAS-2B cells against apoptosis and inflammation.** LINC00612, lncRNA long intergenic non-coding 00612; STAT3, signal transducer and activator of transcription 3; A2M, alpha-2-macroglobulin.

the promoter sequences of A2M in BEAS-2B cells. Indicating that STAT3 could target A2M promoter in BEAS-2B cells (Fig. 6D). Primer 2 was chosen for subsequent study owing to its most powerful enrichment efficiency. As shown in Fig. 6E, knockdown of LINC00612 impaired the binding of p-STAT3 to the A2M promoter, which confirmed that LINC00612 was critical for the binding of p-STAT3 with A2M promoter. Taken together, our data manifested that LINC00612 indeed enhanced the interaction between p-STAT3 and A2M promoter, suggesting that LINC00612, p-STAT3, and A2M promoter may form a complex to regulate A2M transcription.

## DISCUSSION

Increasing evidence suggests that LncRNA is involved in regulating COPD pathogenesis (*De Smet et al., 2015*). Nevertheless, the study of the biological functions of lncRNAs at the stages of occurrence and development of COPD is just beginning, and hence the mechanisms have yet to be fully elucidated. In this article, the results confirmed that LINC00612 can inhibit LPS-induced BEAS-2B inflammation and cell apoptosis by promoting A2M expression. Mechanically, this could be due to the enhancement of A2M transcription by recruiting p-STAT3 to bind to the A2M promoter, as illustrated in Fig. 7.

*Luo et al. (2020)* showed that LINC00612 was decreased in COPD tissues. Combining this finding, our previous study detected that there was a downregulation of LINC00612 in the peripheral blood of COPD sufferers (*Qian et al., 2018*) and we have verified the result

in the present study. BEAS-2B cells are derived from the normal bronchial epithelium of non-cancerous individuals, which retain epithelial cell properties and to a certain extent reflect the state of the body. Epithelial cells play a crucial role in the pathological process of lung inflammation, since they can release inflammatory cytokines. It was commonly used as a model cell line in respiratory biomedical research. Exposure to LPS may stimulate alveolar bronchial epithelial cells and macrophages to secret IL-6, MMPs, and TNF-α (*Pugin et al., 1993*). Under inflammations, the injury and remodeling of the lungs can be triggered by the introduction of inflammatory cells (*Pugin et al., 1993*; *Brooks et al., 2020*), which accelerated the development of COPD. Therefore, LPS was usually adopted to construct an *in vitro* cellular model of COPD. Consistently, our results showed that LINC00612 expression was decreased significantly in LPS-induced BEAS-2B cells. Furthermore, the findings suggested a protective effect of LINC00612 on apoptosis and inflammation cytokines (IL-6 and TNF-α) in LPS-induced BEAS-2B cells, which was in line with the previous studies (*Luo et al., 2020*). These data suggested that LINC00612 participates in the progression of COPD.

LncRNAs have tissue-specific expression patterns and dynamic functions in a variety of cellular circumstances, including decoy miRNAs or RNA binding proteins (RBPs) (*Ransohoff, Wei & Khavari, 2018*). Previous literature had described LINC00612 functions as a miRNA sponge (*Luo et al., 2020*; *Miao et al., 2019*; *Zhou, Li & Yang, 2020*), in other words, the functions were illustrated as an interaction that has typically been thought to occur in the cytoplasm. For example, *Miao et al. (2019)* revealed that LINC00612 enhanced the proliferation and invasion ability of bladder cancer cells by sponging miR-590 to upregulate PHF14 expression. *Zhou, Li & Yang (2020)* indicated that LINC00612 functioned as a ceRNA for miR-214-5p so that the proliferation and intrusion of osteosarcoma had been augmented. However, the role of lncRNA in the gene regulation of human cells is more complex than previously realized. LncRNAs expressed in the nucleus can regulate the transcription of neighbor genes in cis by altering their chromatin states or distant genes in trans by recruiting transcription factors to local locus (*Wang et al., 2019*; *Wang & Chang, 2011*; *Hung & Chang, 2010*; *Kopp & Mendell, 2018*; *Engreitz et al., 2016*). Although the role of LINC00612 in COPD has been identified in a previous study (*Luo et al., 2020*), its mechanism has not been fully elucidated. In contrast to the previous study focused on the mechanism of lncRNA in the cytoplasm, our research paid more attention to the mechanism in the nucleus, and the results showed that LINC00612 regulated the co-expressed coding gene A2M by targeting transcription factors STAT3.

A2M is a protease inhibitor and cytokine transporter. In addition to using bait-and-trap mechanisms to inhibit a broad spectrum of proteases, A2M can also inhibit inflammatory cytokines, including IL-6 (*Nancey et al., 2008*; *Matsuda et al., 1989*) and TNF-α (*Jeng et al., 2011*; *Wollenberg et al., 1991*), thus disrupting inflammatory cascades (*Rehman, Ahsan & Khan, 2013*). We found that the expression of A2M could be downregulated in the peripheral venous blood of COPD patients and the knockdown of A2M could interfere with the inhibition of LINC00612 on LPS-induced inflammation and apoptosis in BEAS-2B cells. Therefore, it was proved that A2M might be potentially able to abate airway inflammation during COPD infection. *Poller, Barth & Voss (1989)* reported a patient with

A2M genetic alteration and partial serum A2M deficiency who had an early onset of lung disease and rapid progression to very severe COPD. This might be related to the A2M gene alteration affecting the region of promoters and causing a decrease in transcription. The synthesis of A2M is primarily regulated at the transcriptional level. *Uskokovic et al. (2007)* found that A2M gene expression relied on the interplay between STAT3 and NF-κB. *Lerner et al. (2003)* indicated the synergy between STAT3 and glucocorticoid receptors for a transcriptional increase of A2M gene. *Yoo et al. (2001)* revealed that sequence-specific DNA and protein interactions at a specific array of DNA elements in the A2M promoter were critical for cooperative transcriptional activation of A2M transcription by STAT3 and c-Jun. All these studies confirmed the STAT3 binding sites in the A2M promoter region in hepatocarcinoma cell lines of rats. Following the above research, our study found that STAT3 was bound to the A2M promoter in BEAS-2B cells.

STAT3, as a transcriptional factor, is a key molecule that mediates inflammatory response affected by many factors, and then encodes a set of proteins to play pro-inflammatory or anti-inflammatory effects (*Yang et al., 2013*). Increasing evidence suggested that by several mechanisms, lncRNA could directly alter the expression and activation of STAT3. For example, lncRNA-p21 can impede the transcriptional activity of STAT3 through a direct connection to STAT3 (*Jin et al., 2019*). LncRNA LEISA recruited STAT3 to bind the promoter of IL-6 and upregulated IL-6 expression (*Wu et al., 2021*). ASRPS is a LINC00908-encoded polypeptide that could directly connect to the CCD domain of STAT3, and thus reduce the phosphorylation of STAT3 (*Wang et al., 2020a*). We found that the knockdown of LINC00612 impaired the connection between p-STAT3 and the A2M promoter, which meant that LINC00612 was critical for the binding of STAT3 to the A2M promoter. Therefore, we speculated that STAT3 is mainly located in the cytoplasm, and after STAT3 was phosphorylated and activated, it translocated to the nucleus and was recruited by LINC00612, where it regulated A2M expression. In addition, we observed that A2M mRNA was significantly enriched in biotin-labeled LINC00612 pull-down samples. We speculate that LINC00612 binds to A2M mRNA after transcription, and this binding may be through the formation of a complex of LINC00612, p-STAT3 and A2M promoters. The limitation of this experiment was the lack of *in vivo* data regarding the effect of LINC00612 on the pathogenesis of COPD by affecting A2M transcription *via* STAT3. The task will be supplemented in our future work. In addition, only one cell line was used in our experiment, which may be insufficient for the results. Future experiments need to be further verified with other cell lines.

An extended understanding of the molecular interaction types in STAT3, LINC00612, and A2M promoter might provide new insights in developing more effective strategies for different biological and clinical contexts. To our knowledge, this research is the first to demonstrate the binding site of STAT3 to the promoter of A2M in human healthy lung epithelial cell lines (BEAS-2B) and reveals that LINC00612 can bind with p-STAT3 and then enhance the genetic transcription of A2M. This finding suggests the existence of a new type of regulation of the expression of A2M in BEAS-2B cells.

## CONCLUSIONS

In summary, we found that LINC00612 is abnormally low expressed in patients with COPD, and may serve as a new diagnostic marker to predict COPD. Additionally, LINC00612 enhances BEAS-2B cells against apoptosis and inflammatory reactions mediated by LPS, whereas the degree of the enhancement was attenuated by A2M knockdown. Mechanistically, LINC00612 promoted the transcription of A2M by recruiting STAT3 to bind to A2M promoter. The results of this study provide a new research idea, theoretical basis and target selection to further investigate COPD pathogenesis. Apparently, a more in-depth investigation is needed to determine whether it can be targeted clinically as a treatment strategy for LINC00612.

## ACKNOWLEDGEMENTS

We thank all the members for their generous participation.

### Funding

This study was supported by grants from the Jiangsu Province Social Development Project (BE2020651 to Qian Zhang), the Jiangsu Province "333 Talents" Project (BRA2020015 to Qian Zhang), the Changzhou Sci & Tech Program (CE20205023 to Qian Zhang), the China Postdoctoral Science Foundation (2020M670011ZX to Zhengdao Mao), the Sci & Tech Project for Young Talents of Changzhou Health Commission (WZ202010 to Yujia Shi) and Changzhou Healthy Seedling Talent Training Project (CZQM2020077 to Yujia Shi). The funders had no role in study design, data collection and analysis, decision to publish, or preparation of the manuscript.

### Grant Disclosures

The following grant information was disclosed by the authors:
Jiangsu Province Social Development Project: BE2020651.
Jiangsu Province "333 Talents" Project: BRA2020015.
Changzhou Science & Technology Program: CE20205023.
China Postdoctoral Science Foundation: 2020M670011ZX.
Science & Technology Project for Young Talents of Changzhou Health Commission: WZ202010.
Changzhou Healthy Seedling Talent Training Project: CZQM2020077.

### Competing Interests

The authors declare that they have no competing interests.

### Author Contributions

- Xinru Xiao conceived and designed the experiments, performed the experiments, prepared figures and/or tables, authored or reviewed drafts of the article, and approved the final draft.

- Wei Cai performed the experiments, prepared figures and/or tables, and approved the final draft.
- Ziqi Ding analyzed the data, prepared figures and/or tables, and approved the final draft.
- Zhengdao Mao analyzed the data, prepared figures and/or tables, and approved the final draft.
- Yujia Shi analyzed the data, prepared figures and/or tables, and approved the final draft.
- Qian Zhang conceived and designed the experiments, authored or reviewed drafts of the article, and approved the final draft.

## Human Ethics

The following information was supplied relating to ethical approvals (*i.e.*, approving body and any reference numbers):

Changzhou Second Hospital granted Ethical approval to carry out the study within its facilities ([2022]KY113-01).

## Data Availability

The raw measurements are available in the Supplemental Files.

## Supplemental Information

Supplemental information for this article can be found online at http://dx.doi.org/10.7717/peerj.14986#supplemental-information.

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
