# Peer review of "LincRNA00612 inhibits apoptosis and inflammation in LPS-induced BEAS-2B cells via enhancing interaction between p-STAT3 and A2M promoter"

_PeerJ, doi:10.7717/peerj.14986_

## Round 0.1 · original submission · Major Revisions

Please attend to the reviewers' comments.

Reviewer 1 ·

Basic reporting

The manuscript was written in a clear and concise manner. The authors should properly cite the catRAPID software by citing the original research paper.

Experimental design

The manuscript by Xiao et al described a role of lincRNA00612 in promoting the interaction between p-STAT3 and A2M promoter in LPS-induced BEAS-2B cells. The authors established the LPS-treated BEAS-2B cells as a model for COPD and carried out the molecular interaction and physiological analysis in this model. The physical interaction between lincRNA00612 and pSTAT3 evidenced by biotinylation pull-down is especially exciting.

Validity of the findings

1. In Figure 1A and B, the authors showed that lincRNA00612 and A2M levels were down about 50% in COPD patients in comparison to control. However, in later LPS induction experiments, they chose 10 ug/ml of LPS, which reduced the expression of lincRNA00612 by more than 70%. It seemed to me that 5 ug/ml treatment aligns more with the patient data.

2. In figure 5C, the authors showed that biotinylated lincRNA also interacts with A2M mRNA. Since they showed later that lincRNA00612 binds to the promoter of A2M, it would be nice if the authors could interpret a bit more on this result. Would this suggest that lincRNA00612 binds to A2M mRNA after its transcription?

Reviewer 2 ·

Basic reporting

The manuscript “LincRNA00612 Inhibits Apoptosis and Inflammation in LPS-Induced BEAS-2B Cells via Enhancing Interaction between p-STAT3 and A2M Promoter” is a research article to investigate the effects of lncRNA00612 in the progression of chronic obstructive pulmonary disease (COPD). The results suggested that lncRNA00612 might play as a key role in the occurrence and development of COPD via Enhancing Interaction between p-STAT3 and A2M Promoter. It might be of interest to the readers who focus on this topic. However, some points need to be addressed in the current version.
(1) The originality/novelty of the study should be addressed significantly.

Experimental design

(2) How did the author isolate/purify miRNA from the extract total RNA sample?

Validity of the findings

(3) Future direction for the study should be described in the Conclusion.

Additional comments

(4) Indicate difference of this work with already published works.
(5) Add one table comparing results of this work with already published works.
(6) Discussion needs improvement.
(7) Conclusion to be modified highlighting major outcomes of the study.
(8) It would be better to provide the whole uncropped images of the original western blots.

Reviewer 3 ·

Basic reporting

This manuscript is clear and unambiguous, professional English used throughout.
However, this article includes unsufficient introduction and background about the BEAS-2B cells used in this paper. Is this cell are good model for LPS treatment ideal model cell line to mimic COPD? What is main reaction of LPS treatment in this cells?
This article provides the professional article structure, tables. However, their figures are somehow not good enough to verify their claims.
1. Figure 3d, the results are not solid that the siNC treatment have much higher expression of A2M compared to vector.
2. Figure 3f, Linc00612 and siA2M treatment is weird to show any meaningful result and conclusion.
3. Figure 5b, 6b, authors should show the whole STAT3 protein expression rather than only p-STAT3.

Experimental design

This is an original primary research. As mentioned in basic reporting, authors should explain or provide the revised data to support their claim.

Validity of the findings

Because of only one cell line used as COPD model in this manuscript, the results are high risk to support their conclusions. Authors should mention in the manuscript or discuss about the limitation of this article.

Additional comments

no

---

## Round 0.2 · accepted · Accept

The authors have addressed all the reviewers' comments. Also, I have reviewed the whole manuscript, and it was improved with the corrections done. In my opinion the manuscript is ready for publication.

Reviewer 1 ·

Basic reporting

All good

Experimental design

All good

Validity of the findings

All good

Additional comments

All good

Reviewer 2 ·

Basic reporting

no comment

Experimental design

no comment

Validity of the findings

no comment

Additional comments

In the revised manuscript of "LincRNA00612 Inhibits Apoptosis and Inflammation in LPS-Induced BEAS-2B Cells via Enhancing Interaction between p-STAT3 and A2M Promoter", the authors have ansewered all the queries raised very well.
Manuscript is acceptable in the revised form. I have no more suggestions.

Reviewer 3 ·

Basic reporting

The authors fixed all of my concerns in the revision.

Experimental design

The revised data are fine to answer the questions.

Validity of the findings

The findings in this manuscript are validated based on their experiments.